# NPvis: An Interactive Visualizer of Peptidic Natural Product–MS/MS Matches

**DOI:** 10.3390/metabo12080706

**Published:** 2022-07-29

**Authors:** Olga Kunyavskaya, Alla Mikheenko, Alexey Gurevich

**Affiliations:** Center for Algorithmic Biotechnology, St. Petersburg State University, 199004 St. Petersburg, Russia; o.kunyavskaya@spbu.ru (O.K.); a.mikheenko@spbu.ru (A.M.)

**Keywords:** peptidic natural products, bioinformatics, mass spectrometry, visualization, software

## Abstract

Peptidic natural products (PNPs) represent a medically important class of secondary metabolites that includes antibiotics, anti-inflammatory and antitumor agents. Advances in tandem mass spectra (MS/MS) acquisition and in silico database search methods have enabled high-throughput PNP discovery. However, the resulting spectra annotations are often error-prone and their validation remains a bottleneck. Here, we present NPvis, a visualizer suitable for the evaluation of PNP–MS/MS matches. The tool interactively maps annotated spectrum peaks to the corresponding PNP fragments and allows researchers to assess the match correctness. NPvis accounts for the wide chemical diversity of PNPs that prevents the use of the existing proteomics visualizers. Moreover, NPvis works even if the exact chemical structure of the matching PNP is unknown. The tool is available online and as a standalone application. We hope that it will benefit the community by streamlining PNP data analysis and validation.

## 1. Introduction

Peptidic natural products (PNPs) consist of ribosomal [1] and nonribosomal peptides [2] and constitute an invaluable source of new biomedical substances [3]. Tandem mass spectrometry (MS/MS) allows for the rapid and cheap scanning of thousands of PNPs [4,5,6,7], but the interpretation of the resulting voluminous data remains challenging. The reference PNP spectral libraries are small, so researchers often rely on in silico identification methods [8,9,10]. These tools enable the search for larger PNP chemical databases, but their outputs are error-prone and require manual validation.

The visualisation of peptide–spectrum matches is a powerful method for validating in silico identifications by annotating MS/MS peaks with peptide fragments. Multiple state-of-the-art proteomics tools address this problem [11,12,13], while the current metabolomics instruments can annotate peaks only with chemical formulas rather than structures of the underlying compound fragments [14]. Despite the similarity between PNPs and peptides, conventional proteomics visualizers are unsuitable for the analysis of PNP–spectrum matches due to the high chemical diversity of the metabolites. Many PNPs have cyclic, branched or more sophisticated structures, contain non-proteinogenic amino acids and complex post-translational modifications and include non-amide linkages of amino acids [15]. Thus, conventional proteomics tools are unsuitable for PNP analysis.

Here, we present NPvis, a versatile interactive visualizer for exploring and validating PNP–spectrum matches. Our tool works with non-linear compounds, supports an extended alphabet of amino acids and handles PNP-specific linkage bonds. NPvis is freely available as a web service and a command-line utility from http://cab.spbu.ru/software/npvis/ (accessed on 15 June 2022). We also integrated the tool with the leading MS/MS metabolomics platforms.

## 2. Results

### 2.1. NPvis Overview

We propose using NPvis as a post-processing tool after applying in silico database search methods to MS/MS datasets. The instrument will help to eliminate wrong hits and select the most reliable annotations among the reported candidate compounds (Appendix A).

NPvis takes, as an input, a targeted or untargeted metabolomics spectrum (MS/MS) and a peptidic natural product (PNP). The tool identifies the breakable bonds in the PNP structure, models its tentative MS/MS fragmentation and annotates the experimental spectrum with the predicted fragments (Section 4). We call an observed MS/MS peak annotated by a fragment if their masses are within a small user-configurable threshold. We recommend setting the threshold to 0.01–0.05 Da for high-resolution MS/MS instruments, such as Orbitrap and Q-TOF, and using 0.5–2 Da for low-resolution facilities, such as Ion Trap and QqQ (the default value is 0.03 Da). The NPvis outputs an interactive report visualising the annotated MS/MS peaks and the corresponding PNP fragments (Figure 1).

Our tool supports two running modes named “PNP” and “PNP with modification”. The former mode assumes that the mass spectrum exactly corresponds to the provided compound. In this case, NPvis attributes the mass difference between the spectrum and the PNP to an instrument measurement error or to the mass of an adduct used for MS/MS ionisation. The latter mode implies that the mass spectrum is related to the provided compound but it corresponds to a modified or mutated variant of that PNP. In this case, NPvis considers the mass difference as the mass of an (unknown) modification and locates its most likely position in the PNP structure. The tool highlights the affected PNP fragment in the interactive report and annotates MS/MS peaks relying on this information (Figure 1d, Section 4).

### 2.2. Web Server Design and Features

We launched the NPvis web server at http://cab.cc.spbu.ru/npvis/ (accessed on 15 June 2022). Figure 1 and Appendix A present the main page overview and visualisation examples in different running modes. The server accepts spectra in the common mass spectrometry formats, such as MGF, mzXML and mzML, and chemical structures as MDL MOL files or text strings in the SMILES format [16]. Users can select the NPvis running mode (“PNP” or “PNP with modification”) and more advanced settings, such as the mass error tolerance for the peaks annotation, the charge and the type of ionization adduct. The resulting visualisations are generated within seconds. To accustom users to input data formats and the NPvis parameters, we provided sample data via the Load Sample Data button and detailed control descriptions available from the Help menu in the top bar (Figure 1).

Users can download NPvis reports as portable HTML files and share them with their peers. We also provided publication-ready PNG images of the annotated spectra; both the entire view or any zoomed area. If the data are sensitive, users can download the command-line version of NPvis from the top bar menu and process everything locally. In case of any problems, the Contact button offers a quick feedback from the developers.

### 2.3. Integration with Metabolomics Platforms

To widen the potential audience of NPvis, we integrated the tool with leading online metabolomics platforms. In particular, we embedded NPvis reports into the Dereplicator/VarQuest [8,17] workflow at the Global Natural Products Social (GNPS) molecular network [18]. With this workflow, researchers can now search their MS/MS data against PNP databases and see the visualisations of the most reliable hits directly in the native GNPS report window (Appendix A).

To enable easy access to billions of natural product mass spectra available at GNPS and other platforms, such as MassBank [19] and MetaboLights [20], the NPvis accepts and resolves metabolomics universal spectra identifiers (USI) [21]. To simplify the future integration of NPvis by third-party developers, our web server supports HTTP GET requests. A valid request contains a spectrum specified via USI, a compound in the SMILES format and the NPvis running parameters. The web server handles incoming requests and generates the NPvis reports in return. The detailed request format specification and working examples are available from the web server Help page.

## 3. Discussion

Metabolomics witnesses a wide application of tandem mass spectrometry (MS/MS) coupled with in silico identification methods. While these approaches facilitate high-throughput metabolites discovery, they often produce error-prone results requiring manual validation. To simplify the process, we created NPvis, a visualizer of metabolite–spectrum matches. The tool is specifically designed for peptidic natural products (PNPs), a medically important class of secondary metabolites. NPvis allows users to interactively map annotated MS/MS peaks to the corresponding PNP fragments and assess the annotation reliability. The tool can annotate a spectrum, even if the corresponding PNP is unknown, and only its structurally related variant is provided.

To the best of our knowledge, NPvis is the first interactive MS/MS viewer suitable for the annotation of PNP fragmentations. In contrast to the existing metabolomics visualizers, it associates MS/MS peaks with structures of the PNP fragments, which is substantially more informative than chemical formula annotations. However, there is still room for improvement and we plan to extend NPvis functionality in future releases. Our tool currently lacks many advanced features of its proteomics counterparts. In particular, NPvis does not annotate MS/MS peaks that occurred due to neutral losses and does not provide manual peak annotation. In addition, the tool is limited to PNPs and cannot handle compounds without a peptidic backbone, such as polyketides and terpenes. The recent creation of MS/MS fragmentation models for general metabolites [22] enables the NPvis extension to compounds with arbitrary chemical structures. Finally, our tool processes only a single compound–spectrum pair per run. We plan to support multi-file input and design-merged NPvis output visualizations so that users can conveniently compare several candidate annotations per spectrum or compound.

We designed NPvis for diverse requests of the community. The console version of the tool fits researchers working with sensitive data, whereas the web version and integration with popular metabolomics platforms suit users without command-line experience. We hope that NPvis will make the tracking of known and discovery of novel PNPs in MS/MS data more robust and descriptive.

## 4. Materials and Methods

### 4.1. Pipeline and Web Server Implementation

The NPvis pipeline was written in Python, the structure fragmentation and MS/MS peaks annotation were implemented in C++, the interactive HTML visualisations were made in JavaScript with the use of the jQuery library and the web server was implemented in Django [23]. The source code of the server and the command-line tool are available at https://github.com/ablab/NPvis (accessed on 15 June 2022).

The pipeline natively accepts spectra in the mzXML and MGF formats; all other common spectra formats are converted to MGF using the msconvert utility from the ProteoWizard package [24]. Spectra provided via USI are resolved using the Metabolomics Spectrum Identifier Resolver [25] and converted from the JSON representation to MGF using the matchms Python library [26]. Chemical structures are natively accepted in the MDL MOL V3000 format; the SMILES representations are converted to MOL files with the RDKit Python library [27].

The NPvis spectra visualisation was implemented on top of the Lorikeet proteomics visualizer (http://uwpr.github.io/Lorikeet/, accessed on 15 June 2022). PNP structures were visualised with the ChemDoodle Web library [28]. The correspondences between the annotated MS/MS peaks and the PNP fragments were generated with the print_score utility from the NPDtools package [8].

### 4.2. MS/MS Fragmentation Model and Identification of Modifications

NPvis utilises the Dereplicator model of the PNP fragmentation in mass spectrometers [8]. Given a chemical structure, the tool first identifies the bonds that likely break in an MS/MS instrument. We refer to them as breakable bonds and include in them amide and other linkage bonds common among PNPs containing heterocycles, such as thiazole and oxazole. Next, NPvis constructs a PNP graph with breakable bonds as edges and everything in between as vertices (amino acids, lipid tails, etc.) Finally, we modeled the tentative MS/MS fragments by splitting the PNP graph into subgraphs by removing every single edge or pair of edges. The latter option is important for non-linear PNPs—which are widespread—since a single edge removal may not break their graphs into two parts (Appendix A).

In the “PNP with modification” mode, NPvis detects the tentative location of the modification using an approach inspired by the VarQuest algorithm [17]. The mass of the (unknown) modification is calculated as the difference between the spectra precursor mass and the mass of the provided PNP. The retrieved mass shift is sequentially applied to each vertex of the PNP graph and the resulting modified PNP is scored against the spectrum using NPScore [29]. The highest scoring compound determines the putative modification position.

## Figures and Tables

**Figure 1 metabolites-12-00706-f001:**
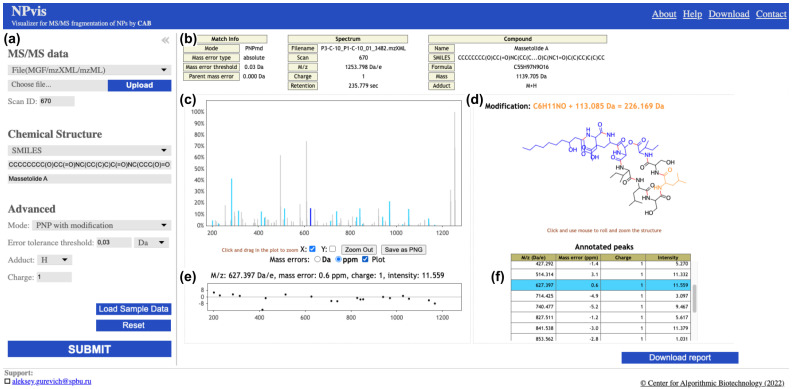
The NPvis web server visualisation of a mass spectrum matching an (unknown) massetolide A variant. (**a**) The foldable control panel allows users to specify the input data and the NPvis settings. (**b**) The metadata panel informs about the provided spectrum and compound and the match parameters. (**c**) The zoomable spectrum panel highlights annotated (sky blue) and not-annotated (grey) MS/MS peaks. (**d**) The zoomable and rotatable compound panel presents the overall chemical structure (black) and the breakable bonds (red) of the compound. In the “PNP with modification” mode, this panel also shows the mass and proposed location of the modification (orange). In this example, the modification is +113.085 Da, which likely corresponds to an insertion of a single leucine or isoleucine residue (monoisotopic mass 113.084 Da). A user may click on an annotated peak in the spectrum panel to see the corresponding structure fragment in the compound panel (both colored blue). (**e**) The hideable mass error plot of absolute (Da) and relative (ppm) discrepancies between the masses of the annotated peaks and the corresponding structure fragments. (**f**) The scrollable list of annotated peaks highlights the selected peak (if any). The (**b**–**f**) panels are present in the downloadable interactive reports (the bottom-right button), as well as reports generated by command-line NPvis.

## Data Availability

NPvis source code and command-line release are available from GitHub at http://github.com/ablab/npvis (accessed on 15 June 2022). NPvis web server is available at http://cab.cc.spbu.ru/npvis/ (accessed on 15 June 2022). Interactive standalone examples of NPvis visualizations are available at http://cab.spbu.ru/software/npvis/ (accessed on 15 June 2022). Examples of the GNPS-integrated NPvis visualization are at https://gnps.ucsd.edu/ProteoSAFe/result.jsp?task=49e24d19c2b24bc7a82cd65319ce91c0&view=view_significant (accessed on 15 June 2022).

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
