# Peer review of "NPvis: An Interactive Visualizer of Peptidic Natural Product–MS/MS Matches"

_metabolites, 2022, doi:10.3390/metabo12080706_

Round 1
Reviewer 1 Report
This article provided a visualizer for peptidic natural products (PNP), to explore and validate PNP-spectrum matches. The authors provided a figure for their visualizer and designed some integration with metabolic platforms. In general, the strengths of their visualizer compared with others can be provided. Besides, the matching method for PNP is missing in the manuscript, which is the core of NPvis. Below are the detailed suggestions and questions.
1. The definition of PNP (represents a medically-important class of secondary metabolites) means that PNP belongs to metabolites. The authors compared it with proteomic tools in the introduction, while the comparison with metabolic tools was missing.
2. How did the authors solve the error-prone problem compared with the conventional in silico identification method?
3. What is the matching algorithm used in NPvis? Is this method specially designed for NPvis and targeted at PNP, which has cyclic, branched, or more sophisticated structures, contains non-proteinogenic amino acids and complex post-translational modifications, and includes non-amide linkages of amino acids?
4. Adances on mass spectrometry (VIEW 2020, 1:20200063; Angew.Chem.Int.Ed. 2021, 60, 12504; Angew.Chem.Int.Ed. 2021, 60, 11310) and machine learning (Angew.Chem.Int.Ed. 2020, 59, 10831; Nature Communications 2020, 11, 3556; Advanced Science 2020, 7, 2002021) should be included.
5. The novelty of this manuscript should be enhanced on the interactive visualizer. For example, what is the comparison with other metabolic visualizers? What are NPvis’ strengths compared with other metabolic visualizers? Is there any other novelties of NPvis?
Reviewer 2 Report
please see attached file

Round 2
Reviewer 1 Report
The paper can be accepted.